# Synergistic Neuroprotection Through Epigenetic Modulation by Combined Curcumin-Enriched Turmeric Extract and L-Ascorbic Acid in Oxidative Stress-Induced SH-SY5Y Cell Damage

**DOI:** 10.3390/foods14050892

**Published:** 2025-03-05

**Authors:** Jurairat Khongrum, Nootchanat Mairuae, Tongjit Thanchomnang, Man Zhang, Gang Bai, Nut Palachai

**Affiliations:** 1Multidisciplinary Research Institute, Chiang Mai University, Chiang Mai 50200, Thailand; jurairat.kh@cmu.ac.th; 2Functional Food Research Center for Well-Being, Multidisciplinary Research Institute, Chiang Mai University, Chiang Mai 50200, Thailand; 3Biomedical Research Unit, Faculty of Medicine, Mahasarakham University, Mahasarakham 44000, Thailand; nootchanat.m@msu.ac.th (N.M.); tongjit.t@msu.ac.th (T.T.); 4State Key Laboratory of Medicinal Chemical Biology, College of Pharmacy and Tianjin Key Laboratory of Molecular Drug Research, Nankai University, Tianjin 300353, China; manzhang@nankai.edu.cn (M.Z.); gangbai@nankai.edu.cn (G.B.)

**Keywords:** curcumin, turmeric, L-ascorbic acid, neurodegenerative diseases, epigenetics

## Abstract

Epigenetic modulation plays a crucial role in neuroprotection by regulating cellular responses to stress, inflammation, and oxidative damage, particularly in neurodegenerative diseases. Recognizing the therapeutic potential of epigenetic regulators, this study investigated the synergistic neuroprotective effects of curcumin-enriched turmeric extract combined with L-ascorbic acid, focusing on its modulation of epigenetic pathways in oxidative stress-induced neuronal damage. SH-SY5Y neuronal cells were treated with the combination at 20 and 40 µg/mL, and subsequently exposed to 200 µM hydrogen peroxide (H_2_O_2_) to induce oxidative stress. Cell viability was assessed using the MTT assay, while neuroprotective mechanisms were evaluated by analyzing the markers of epigenetic modulation, oxidative stress, inflammation, and apoptosis. The combination significantly enhanced cell viability, upregulated sirtuin-1 (SIRT1), and reduced DNA methyltransferase 1 (DNMT1) expression, indicating effective epigenetic modulation. Enhanced antioxidant defenses were observed, as evidenced by increased activities of superoxide dismutase (SOD), catalase (CAT), and glutathione peroxidase (GSH-Px), along with decreased malondialdehyde (MDA) and reactive oxygen species (ROS) levels, alleviating oxidative stress. Additionally, it suppressed nuclear factor kappa B (NF-κB) activity and its downstream mediator interleukin-6 (IL-6), thereby mitigating inflammation. The treatment also increased anti-apoptotic Bcl-2 expression while reducing pro-apoptotic markers, including caspase-3 and caspase-9, suggesting inhibition of the intrinsic apoptotic pathway. These findings highlight the novel neuroprotective effects of this combination, demonstrating its ability to modulate epigenetic pathways while reducing oxidative stress, suppressing inflammation, and preventing undesired apoptosis. Its multifaceted neuroprotective properties make it a promising functional ingredient in functional foods for neurodegenerative disease intervention. However, further investigations, including animal studies and clinical trials, are essential to evaluate its safety and therapeutic potential.

## 1. Introduction

Neurodegenerative diseases, including Alzheimer’s disease (AD), Huntington’s disease (HD), Parkinson’s disease (PD), and multiple-system atrophy (MSA), are characterized by the progressive degeneration of neurons, leading to impairments in cognition, motor function, and overall quality of life [1]. Despite decades of research, current therapeutic approaches primarily focus on symptom management rather than addressing the underlying causes of these conditions [2]. This underscores a critical gap in the development of treatments that target the fundamental mechanisms driving neurodegeneration. Among these mechanisms, the epigenetic regulation of oxidative stress, inflammation, and neuronal apoptosis has emerged as a pivotal yet underexplored area of research.

Epigenetic modifications, including DNA methylation, histone modifications, and non-coding RNA regulation, influence gene expression without altering the DNA sequence. The dysregulation of these processes disrupts antioxidant gene expression, leading to an accumulation of ROS and heightened oxidative stress. For example, the hypermethylation of genes encoding antioxidant enzymes such as SOD and GSH-Px diminishes their activity, exacerbating oxidative damage and neuronal loss [3,4,5].

Inflammation, a hallmark of neurodegenerative diseases, is similarly governed by epigenetic mechanisms. The activation of microglia and the release of pro-inflammatory cytokines, including IL-6 and tumor necrosis factor-alpha (TNF-α), are amplified by aberrant histone acetylation and methylation. Epigenetic regulators such as sirtuins, particularly SIRT1, play a crucial role in suppressing NF-κB signaling, a key inflammatory pathway. Dysregulated sirtuin activity contributes to chronic neuroinflammation, further accelerating neuronal dysfunction [6,7].

In addition, epigenetic dysregulation impairs genes essential for neuronal survival, synaptic plasticity, and neurotransmitter production. In HD, for instance, hypermethylation of the brain-derived neurotrophic factor (BDNF) gene compromises synaptic integrity and neuronal survival [8]. Similarly, epigenetic alterations in dopamine-related pathways contribute to the loss of dopaminergic neurons in PD. The disruption of long non-coding RNAs (lncRNAs) and microRNAs (miRNAs) further exacerbates neuronal signaling deficits, intensifying neurodegeneration [9].

These interconnected epigenetic changes create a feedback loop that perpetuates oxidative stress, inflammation, and neuronal apoptosis. While these processes have been extensively studied individually, a significant knowledge gap remains in understanding how their integrated modulation could mitigate neurodegenerative progression. Targeting these pathways offers a promising therapeutic avenue to restore cellular homeostasis and slow disease progression.

Environmental and lifestyle factors significantly influence epigenetic regulation, with diet being a key modulator. Beyond sustaining physiological functions, dietary components profoundly impact gene expression through epigenetic pathways [10]. Plant-based foods, in particular, have demonstrated potential in modulating epigenetic mechanisms [11]. Curcumin, a polyphenol extracted from turmeric, has been shown to regulate oxidative stress, inflammation, and apoptosis via epigenetic modulation [12,13]. Similarly, L-ascorbic acid, abundant in fruits and vegetables, exhibits neuroprotective effects by targeting oxidative and inflammatory pathways [14,15]. However, the precise interplay of these compounds in epigenetic regulation related to neurodegeneration remains poorly characterized.

Traditional Ayurvedic and Asian medicine emphasize the combination of bioactive compounds from multiple natural sources in polyherbal formulations. This synergistic approach aligns with the concept of polytherapy, wherein the simultaneous targeting of multiple pathways enhances therapeutic efficacy [16,17,18]. The combined use of curcumin and L-ascorbic acid shows significant promise for addressing neurodegenerative conditions by effectively modulating the epigenetic pathways to counteract oxidative damage, inflammation, and neuronal apoptosis.

Building on these insights, we hypothesized that curcumin-enriched turmeric extract combined with L-ascorbic acid could mitigate neuronal damage and improve cell survival in SH-SY5Y neuroblastoma cells subjected to H_2_O_2_-induced oxidative stress.

To evaluate this hypothesis, this study investigated the modulation of epigenetic pathways, particularly focusing on SIRT1 and DNMT1. Biomarkers linked to oxidative stress (ROS and MDA), antioxidant defenses (SOD, CAT, and GSH-Px), inflammation (NF-κB and IL-6), and apoptosis (Bcl-2, caspase-3, and caspase-9) were assessed. These evaluations aimed to provide a deeper insight into how this combination may offer neuroprotective effects by influencing these critical biological processes, thereby supporting the potential of functional ingredients in the development of functional foods for mitigating neurodegenerative conditions.

## 2. Materials and Methods

### 2.1. Test Substances Used in the Study

Curcumin from *Curcuma longa* (turmeric) rhizome, with a purity of approximately 65%, as confirmed by high-performance liquid chromatography (HPLC) analysis, was obtained from Sigma-Aldrich (MilliporeSigma, St. Louis, MO, USA) (CAS Number: 458-37-7). Likewise, L-ascorbic acid (vitamin C), with a purity of ≥99.0% as determined by HPLC analysis, was sourced from Sigma-Aldrich (MilliporeSigma, St. Louis, MO, USA) (CAS Number: 50-81-7). These compounds, known for their neuroprotective properties, were combined in this study to evaluate their synergistic effects.

A 1:1 ratio of curcumin to L-ascorbic acid was chosen to optimize their interaction and enhance their neuroprotective effects. This balanced ratio was selected to allow both compounds to effectively modulate epigenetic pathways, oxidative stress, inflammation, and apoptosis. By maintaining adequate concentrations of each compound, this approach aims to achieve synergistic effects while minimizing the potential risk of toxicity that could arise from higher doses of either compound when used alone.

### 2.2. Evaluation of Biological Activities

The biological activities of the samples were analyzed through a series of assays to assess their antioxidant and anti-inflammatory properties.

#### 2.2.1. DPPH Assay for Free Radical Scavenging

The DPPH assay was used to evaluate the antioxidant activity of the samples by measuring their ability to neutralize 1,1-diphenyl-2-picrylhydrazyl (DPPH) radicals. A 0.3 mL aliquot of the sample, prepared at concentrations ranging from 0 to 1000 µg/mL, was mixed with 2 mL of a 0.1 mM DPPH solution in methanol. The mixture was thoroughly vortexed and incubated at room temperature (25 °C) for 30 min. Absorbance was measured at 517 nm using a BioTek Synergy H1 Multimode Microplate Reader (BioTek Instruments, Winooski, VT, USA). Methanol served as the blank control. The half-maximal effective concentration (EC50), representing the sample concentration required to inhibit 50% of DPPH radicals, was calculated and reported [19,20,21]. 

#### 2.2.2. FRAP Assay: Ferric Reducing Antioxidant Power

The ferric reducing antioxidant power (FRAP) assay was used to evaluate the samples’ ability to reduce ferric tripyridyltriazine (Fe^3^⁺-TPTZ) to its ferrous form (Fe^2^⁺-TPTZ). A freshly prepared FRAP reagent was made by mixing 20 mM ferric chloride (FeCl₃), 300 mM acetate buffer, and 10 mM 2,4,6-tripyridyl-s-triazine (TPTZ) in a 1:10:1 ratio. For the assay, 10 µL of the sample was combined with 190 µL of the FRAP reagent and incubated at 37 °C for 10 min. Absorbance was measured at 593 nm using a BioTek Synergy H1 Multimode Microplate Reader (BioTek Instruments, Winooski, VT, USA). The EC50 value was used to quantify and express the samples’ antioxidant capability [19,20,21]. 

#### 2.2.3. Radical Scavenging Assay for ABTS

The ABTS•⁺ assay was performed to evaluate the ability of samples to scavenge 2,2-azinobis (3-ethylbenzothiazoline-6-sulfonic acid) (ABTS) radicals. To generate the ABTS radical cation (ABTS•⁺), 7 mM ABTS was mixed with 2.45 mM potassium persulfate (K_2_S_2_O_8_) in a 2:3 ratio and allowed to react in the dark for 12–16 h. For the assay, 30 µL of the sample at varying concentrations was combined with 120 µL of distilled water, 30 µL of ethanol, and 3 mL of the ABTS•⁺ solution. The absorbance was measured at 734 nm using a Pharmacia LKB-Biochrom 4060 Spectrophotometer (Biochrom Ltd., Cambridge, UK). The EC50, representing the scavenging activity, was calculated [19,20,21].

#### 2.2.4. Assay for Cyclooxygenase-2 (COX-2) Inhibition

The anti-inflammatory properties of the samples were evaluated using a COX-2 inhibition assay. The assay was performed following the manufacturer’s instructions using a colorimetric COX-2 inhibitor screening kit. The COX-2 enzyme was dissolved in 100 mM Tris-hydrochloric acid (Tris-HCl) buffer (pH 8.0) at a 1:100 ratio to prepare the working solution.

For the assay, 50 µL of the assay buffer, 10 µL of the sample, 10 µL of heme, 10 µL of the COX-2 solution, 20 µL of 10 µM N,N,N’,N’-tetramethyl-*p*-phenylenediamine dihydrochloride (TMPD), and 20 µL of 100 µM arachidonic acid were combined. The mixture was transferred to a 96-well plate and incubated for 30 min at 25 °C. Absorbance was measured at 590 nm using a BioTek Synergy H1 Multimode Microplate Reader (BioTek Instruments, Winooski, VT, USA). The results are expressed as the EC50, representing the concentration required to inhibit 50% of COX-2 activity [19,20,21]. 

### 2.3. Evaluation of the Combination Index (CI) and Dose Reduction Index (DRI)

The CI and DRI were calculated to evaluate the synergistic interactions between curcumin-enriched turmeric extract and L-ascorbic acid.

#### 2.3.1. Calculating the Combination Index

The following formula was used to determine the combination index: CI = (D1/Dx1) + (D2/Dx2)
where Dx1 and Dx2 are the amounts of curcumin-enriched turmeric extract and L-ascorbic acid, respectively. The concentrations of each component that produce the same result when employed separately are denoted by D1 and D2.

The following is how the CI values were interpreted: CI > 1.45: antagonistic effect, CI = 1: additive effect, and CI < 1: synergistic effect [19].

#### 2.3.2. Calculating the Dose Reduction Index

The dose reduction index (DRI) represents the fold reduction in the dose of each component in a combination compared to the dose required for the same effect when the compound is administered alone. The DRI was calculated using the following formula:DRI = Dx1/D1
where Dx1 is the concentration of the compound when combined with another component to produce a half-maximal response, and D1 is the concentration of the single substance required to achieve the same effect independently [19].

These indices provided quantitative insights into the potential synergistic effects of curcumin-enriched turmeric extract and L-ascorbic acid in enhancing neuroprotection. 

### 2.4. Cell Culture

The SH-SY5Y cell line, derived from human neuroblastoma and characterized by neuronal properties, was obtained from the American Type Culture Collection (ATCC, catalog number CRL-2266, Manassas, VA, USA). Cells were cultured in Dulbecco’s modified eagle medium (DMEM; Gibco, Waltham, MA, USA) supplemented with 10% fetal bovine serum (FBS), 1% penicillin–streptomycin, and 1% non-essential amino acids. Cultures were maintained at 37 °C in a humidified incubator with 5% CO_2_. Before each experiment, the culture medium was aspirated and replaced with fresh medium containing H_2_O_2_, with or without the designated treatments [19,20,21].

### 2.5. Cell Viability Assay

The MTT assay was performed to assess the cytotoxic effects of H_2_O_2_ and the curcumin-enriched turmeric extract combined with L-ascorbic acid. SH-SY5Y cells were seeded at a density of 10,000 cells per well in 96-well plates and maintained under standard culture conditions. Cells were then exposed to varying concentrations of H_2_O_2_ (0–800 µM) or the curcumin-enriched turmeric extract combined with L-ascorbic acid (0, 5, 10, 20, 40, 80, 160, 320, 640, and 1280 µg/mL) in serum-free DMEM for 24 h. The optimal H_2_O_2_ concentration for inducing cytotoxicity was selected based on our previous studies [19]. 

The curcumin-enriched turmeric extract and L-ascorbic acid combination was pre-treated for 24 h on SH-SY5Y cells to evaluate its neuroprotective effects against H_2_O_2_-induced damage. After treatment, the medium was replaced with one containing H_2_O_2_, with or without the test compounds. The plates were incubated for 1 h at 37 °C in a humidified 5% CO_2_ atmosphere. Following incubation, 0.5 mg/mL of the MTT reagent (MilliporeSigma, St. Louis, MO, USA) was added to each well. After a 24 h incubation, the MTT solution was discarded, and 100 µL of dimethyl sulfoxide (DMSO) was added to dissolve the formazan crystals. Absorbance was measured at 570 nm using a BioTek Synergy H1 Multimode Microplate Reader (BioTek Instruments, Winooski, VT, USA) [19,20,21].

### 2.6. Determination of Oxidative Stress Markers

#### 2.6.1. Malondialdehyde (MDA)

A modified thiobarbituric acid-reactive substances (TBARS) assay was employed to measure the MDA levels, an indicator of lipid peroxidation. The reaction mixture consisted of 50 µL of 8.1% sodium dodecyl sulfate (SDS), 375 µL of 0.8% thiobarbituric acid (TBA), 375 µL of 20% acetic acid, and 150 µL of distilled water, which were mixed with 50 µL of each sample. All chemicals were sourced from Sigma-Aldrich (MilliporeSigma, St. Louis, MO, USA). The mixture was heated at 95 °C for 60 min, facilitating the interaction between MDA and TBA to form a pink MDA–TBA complex. After cooling the reaction mixture under running tap water, 1250 µL of a 15:1 *v*/*v* n-butanol and pyridine solution (Merck KGaA, Darmstadt, Germany) was added to extract the reaction product. The samples were then centrifuged at 4000 rpm for 10 min, and the top organic layer was carefully collected. Absorbance of the MDA–TBA complex was measured at 532 nm using a Pharmacia LKB-Biochrom 4060 Spectrophotometer (Biochrom Ltd., Cambridge, UK). MDA concentrations were determined using a standard curve prepared from 1,1,3,3-tetramethoxypropane (TMP) Sigma-Aldrich (MilliporeSigma, St. Louis, MO, USA), and expressed as nanograms of MDA per milligram of protein [22,23,24,25]. 

#### 2.6.2. Intracellular Reactive Oxygen Species (ROS)

Intracellular ROS generation was measured using the fluorescent probe 5-(and-6)-carboxy-2′,7′-dichlorofluorescein diacetate (CM-H_2_DCFDA), which is selectively oxidized by ROS. SH-SY5Y cells were plated into 96-well plates and cultured according to standard procedures. After 24 h of incubation, the cells were treated with or without the test compounds. Subsequently, the cells were incubated with 10 µM CM-H_2_DCFDA for 30 min at 37 °C in a CO_2_ incubator, ensuring minimal light exposure to prevent photobleaching. Following a 24 h treatment with H_2_O_2_ in a serum-free medium, the cells were washed with phosphate-buffered saline (PBS). Fluorescence measurement was performed using a BioTek Synergy H1 Multimode Microplate Reader (BioTek Instruments, Winooski, VT, USA) to detect 2′,7′-dichlorofluorescein (DCF), the fluorescent product formed when CM-H_2_DCFDA is oxidized by ROS. The excitation wavelength was set to 488 nm, and the emission wavelength was 520 nm. ROS levels were quantified by measuring the fluorescence intensity of DCF, which reflects the degree of oxidative stress and the antioxidant efficacy of the test compounds [20,21].

By analyzing both MDA levels and ROS generation, this study provided a comprehensive assessment of oxidative stress in SH-SY5Y cells and the protective effects of the combination against oxidative damage.

### 2.7. Determination of Antioxidant Enzyme Activities

#### 2.7.1. Catalase (CAT) Activity

The activity of CAT was measured based on its ability to degrade hydrogen peroxide (H_2_O_2_). A 10 µL aliquot of the sample was mixed with 150 µL of 5 mM potassium permanganate (KMnO_4_), 50 µL of 30 mM H_2_O_2_ (prepared in 50 mM phosphate buffer, pH 7.0), and 25 µL of 4 M sulfuric acid (H_2_SO_4_). The mixture was thoroughly vortexed, and absorbance at 490 nm was measured using a BioTek Synergy H1 Multimode Microplate Reader (BioTek Instruments, Winooski, VT, USA) to monitor the reaction. CAT from Sigma-Aldrich (MilliporeSigma, St. Louis, MO, USA) was used as a reference standard at concentrations ranging from 0 to 100 units/mL. The results are expressed as units per milligram of protein [22,23,24,25].

#### 2.7.2. Superoxide Dismutase (SOD) Activity

SOD activity was measured based on the reduction of superoxide anions (O_2_⁻) generated by xanthine oxidase (XO). The reaction mixture consisted of 0.5 mM xanthine, 0.2 M phosphate buffer (pH 7.8), 0.01 M ethylenediaminetetraacetic acid (EDTA), and 15 mM cytochrome C, combined in a volume ratio of 50:25:1:1 (*v*/*v*). A 200 µL aliquot of the prepared mixture was added to 20 µL of the sample and 20 µL of xanthine oxidase (0.90 mU/mL). SOD activity was determined by measuring absorbance at 415 nm using a BioTek Synergy H1 Multimode Microplate Reader (BioTek Instruments, Winooski, VT, USA). SOD enzyme (MilliporeSigma, St. Louis, MO, USA) was used as a reference standard at concentrations ranging from 0 to 25 units/mL. The results are expressed as units per milligram of protein [22,23,24,25].

#### 2.7.3. Glutathione Peroxidase (GSH-Px) Activity

GSH-Px activity was assessed by using glutathione (GSH) as a reducing agent to catalyze the reduction of H_2_O_2_ and organic peroxides, such as lipid hydroperoxides. The reaction mixture was prepared by combining 10 µL of 1 mM dithiothreitol (DTT), 10 mM monosodium phosphate in distilled water, 100 µL of 40 mM potassium phosphate buffer (KH_2_PO_4_, pH 7.0), 10 µL of 50 mM glutathione, and 100 µL of 30% H_2_O_2_. This mixture was then added to 20 µL of the sample to evaluate GSH-Px activity. After a 10 min incubation at 25 °C, 10 µL of 10 mM 5,5′-dithiobis-(2-nitrobenzoic acid) (DTNB) was introduced. The absorbance of the resulting solution was measured at 412 nm using a BioTek Synergy H1 Multimode Microplate Reader (BioTek Instruments, Winooski, VT, USA). GSH-Px enzyme (MilliporeSigma, St. Louis, MO, USA), with concentrations ranging from 0 to 5 units/mL, served as the reference standard. The results were expressed as units per milligram of protein [22,23,24,25]. 

### 2.8. Western Blotting Analysis

Neuronal protein extraction reagent (N PERTM) lysis buffer (Thermo Fisher Scientific, Inc., Waltham, MA, USA) was used to homogenize the cells before they were lysed. A variety of salts and detergents are included in this buffer to effectively extract proteins from neural cells. Samples were lysed, centrifuged for 10 min at 4 °C at 10,000× *g*, and the supernatant was gathered. A Thermo Scientific NanoDrop 2000c spectrophotometer (Thermo Fisher Scientific, Wilmington, DE, USA) was used to measure the quantities of proteins. 

Tris-glycine sodium dodecyl sulfate (SDS)-polyacrylamide gel electrophoresis (PAGE) loading buffer was combined with 60 µg of protein for Western blotting, and the mixture was denatured by heating it to 95 °C for 5 min. Proteins were moved to a polyvinylidene difluoride (PVDF) membrane after being separated on an SDS-polyacrylamide gel. To avoid non-specific binding, the membrane was incubated for one hour at room temperature with a blocking buffer (5% skim milk in 0.1% Tris-buffered saline with Tween 20 [TBS-T]). 

After blocking, the membrane was incubated overnight at 4 °C with gentle shaking using the following primary antibodies: anti-SIRT1 (no. 2310S), anti-DNMT1 (no. 5032S), anti-NF-κB (no. 8242S), anti-IL-6 (no. 12153S), and anti-caspase-9 (no. 9502S) (Cell Signaling Technology, Danvers, MA, USA; 1:1000); anti-Bcl-2 (no. BF0103) and anti-caspase-3 (no. DF6879) (Affinity Biosciences, Changzhou, Jiangsu, China; 1:1000); and anti-β-actin (no. 4967S) (Cell Signaling Technology, Danvers, MA, USA; 1:2000). Following primary antibody incubation, the membrane was washed with 0.05% TBS-T and subsequently incubated at room temperature with a secondary antibody (Cell Signaling Technology, Danvers, MA, USA; 1:2000) conjugated to anti-rabbit immunoglobulin G (IgG) and horseradish peroxidase (HRP).

Protein bands were detected using the Immobilon Forte Western HRP Substrate (Cat. No. WBLUF0100, Merck KGaA, Darmstadt, Germany). Imaging was performed with the ChemiDoc™ MP system, and band intensities were analyzed using Image Lab software (version 6.0.0, build 25; Bio-Rad Laboratories, Inc., Hercules, CA, USA). Data were normalized to β-actin levels and expressed relative to the control group. The full-length blots are provided in the Appendix A, while representative images were cropped to display the relevant bands [19,20,21,22,23,24,25].

### 2.9. Statistical Analysis

Data are expressed as the mean ± standard error of the mean (SEM). Statistical significance was determined using Tukey’s post hoc test following a one-way analysis of variance (ANOVA) for comparisons between multiple groups. For two-group comparisons, Student’s *t*-test was applied. A *p*-value of less than 0.05 was considered statistically significant. All statistical analyses were performed using SPSS software (version 21.0, IBM Corp., Armonk, NY, USA).

## 3. Results

### 3.1. Synergistic Effects of Curcumin-Enriched Turmeric Extract and L-Ascorbic Acid

This study evaluated the combined effects of curcumin-enriched turmeric extract and L-ascorbic acid to assess the potential synergistic benefits. Table 1 summarizes the findings related to the biological activities of the combination, focusing on antioxidant properties and COX-2 suppression capabilities. The results revealed that the combination exhibited significantly superior antioxidant activities compared to curcumin-enriched turmeric extract alone, as measured by the DPPH, FRAP, and ABTS assays (*p* < 0.001 for all). It also outperformed L-ascorbic acid alone in these assays (*p* < 0.001 for DPPH, *p* = 0.05 for FRAP, and *p* < 0.001 for ABTS). Furthermore, the combination demonstrated the highest COX-2 suppression activity, surpassing the effects of each compound individually (*p* < 0.001 for all).

To quantify the synergistic interaction between the two components, combination indices were calculated and are summarized in Table 2. The combination indices for antioxidant effects, measured by the DPPH, FRAP, and ABTS assays, were 0.41 ± 0.05, 0.55 ± 0.02, and 0.54 ± 0.06, respectively. For COX-2 suppression, the index was 0.71 ± 0.00. All CI values less than 1 suggest a synergistic interaction between curcumin-enriched turmeric extract and L-ascorbic acid.

Moreover, dose reduction indices were calculated to evaluate the reduction in the required doses when the two substances are used in combination. For antioxidant activities, including DPPH, FRAP, and ABTS, the dose reduction indices were 15.59 ± 1.68, 6.20 ± 0.73, and 9.88 ± 0.94 for curcumin-enriched turmeric extract, and 3.20 ± 0.35, 2.67 ± 0.00, and 2.46 ± 0.25 for L-ascorbic acid. For COX-2 suppression, the indices were 2.30 ± 0.00 for curcumin-enriched turmeric extract and 3.58 ± 0.00 for L-ascorbic acid.

Overall, the data demonstrate that the combination of curcumin-enriched turmeric extract and L-ascorbic acid results in synergistic effects across various assays, with combination indices ranging from 0.41 ± 0.05 to 0.71 ± 0.00, and dose reduction indices exceeding 1. This suggests that the combination not only enhances biological activities, but also reduces the effective dose required to achieve the desired effect. 

### 3.2. Hydrogen Peroxide (H_2_O_2_)-Induced Oxidative Stress Model

Oxidative stress induced by H_2_O_2_ is a known contributor to cellular damage in SH-SY5Y cells, disrupting antioxidant defenses and promoting inflammation and apoptosis. This study evaluated the cytoprotective effects of curcumin-enriched turmeric extract combined with L-ascorbic acid using an H_2_O_2_-induced oxidative stress model.

To establish an effective H_2_O_2_ concentration for inducing oxidative stress, SH-SY5Y cells were exposed to varying doses (0–800 µM) for 24 h. Preliminary findings identified 200 µM H_2_O_2_ as an optimal dose to induce oxidative damage while preserving cell viability, making it suitable for further analyses [19].

### 3.3. Cytotoxicity of Curcumin-Enriched Turmeric Extract Combined with L-Ascorbic Acid

The cytotoxicity of curcumin-enriched turmeric extract and L-ascorbic acid was assessed by treating SH-SY5Y cells with concentrations ranging from 0 to 1280 µg/mL for 24 h. As illustrated in Figure 1, cell viability declined with increasing concentrations, from 99.84 ± 1.47% at 5 µg/mL to 51.03 ± 2.03% at 1280 µg/mL. Significant cytotoxic effects were observed at 80 µg/mL and higher (*p* < 0.05 at 80 µg/mL; *p* < 0.01 at 160 µg/mL; *p* < 0.001 at 320 µg/mL and above).

Doses of 20 µg/mL and 40 µg/mL exhibited minimal cytotoxicity while maintaining cell viability. These concentrations were selected for further experiments to explore their potential in mitigating oxidative damage induced by H_2_O_2_.

### 3.4. Neuroprotective Effects of Curcumin-Enriched Turmeric Extract Combined with L-Ascorbic Acid on H_2_O_2_-Induced Oxidative Damage in SH-SY5Y Cell

Figure 2 presents the neuroprotective effects of curcumin-enriched turmeric extract combined with L-ascorbic acid against H_2_O_2_-induced damage in SH-SY5Y cells. Exposure to H_2_O_2_ significantly reduced cell viability compared to the untreated control group (*p* < 0.001). However, treatment with the combination at all tested concentrations markedly improved cell viability. Statistical analysis confirmed that the combination effectively mitigated the cytotoxic effects of H_2_O_2_ (*p* < 0.001 for all doses), supporting its potential neuroprotective properties. 

### 3.5. Epigenetic Modulation by Curcumin-Enriched Turmeric Extract Combined with L-Ascorbic Acid

SIRT1, an NAD⁺-dependent histone deacetylase, and DNMT1, an enzyme responsible for DNA methylation at CpG sites, play key roles in regulating oxidative stress, inflammation, and cellular homeostasis, particularly in neurodegenerative conditions. Therefore, their epigenetic modulation was investigated in this study.

As shown in Figure 3, SH-SY5Y cells exposed to H_2_O_2_ and vehicle exhibited a significant decrease in SIRT1 expression and an increase in DNMT1 expression compared to the control group (*p* < 0.001). Treatment with a combination of curcumin-enriched turmeric extract and L-ascorbic acid significantly upregulated SIRT1 expression compared to the H_2_O_2_ and vehicle group (*p* < 0.001 for all doses). Additionally, the combination treatment, particularly at 40 µg/mL, significantly suppressed DNMT1 expression (*p* < 0.001). While the lower dose (20 µg/mL) did not reach statistical significance, a decreasing trend in DNMT1 expression was observed compared to the H_2_O_2_ and vehicle-treated group.

These findings suggest that the combination exerts a dose-dependent effect on DNMT1 modulation, underscoring its potential role in neuroprotective epigenetic regulation. 

### 3.6. Effects of Curcumin-Enriched Turmeric Extract Combined with L-Ascorbic Acid on Oxidative Stress Markers

Oxidative stress arises from an imbalance between ROS production and antioxidant defenses. MDA, a marker of lipid peroxidation, reflects oxidative damage, while elevated ROS levels indicate heightened oxidative stress. Antioxidant enzymes, including CAT, SOD, and GSH-Px, play essential roles in neutralizing ROS and mitigating cellular damage. This study assessed all these parameters.

Figure 4 and Figure 5 illustrate the effects of curcumin-enriched turmeric extract combined with L-ascorbic acid on ROS and MDA levels, respectively. SH-SY5Y cells exposed to H_2_O_2_ and treated with the vehicle showed significantly elevated ROS and MDA levels compared to the control group (*p* < 0.001 for ROS; *p* < 0.01 for MDA). However, treatment with the combination significantly reduced both ROS and MDA levels (*p* < 0.001 for all doses for ROS; *p* < 0.05 for all doses for MDA) compared to the H_2_O_2_-only group.

Additionally, Figure 6 presents the activities of key antioxidant enzymes, including CAT, SOD, and GSH-Px. Cells treated with H_2_O_2_ and the vehicle demonstrated significantly reduced enzyme activities (*p* < 0.001 for all) compared to the control group. In contrast, cells treated with the higher dose of the curcumin-enriched turmeric extract and L-ascorbic acid combination exhibited a significant increase in enzyme activities (*p* < 0.001 for all) compared to the H_2_O_2_ and vehicle group. Furthermore, treatment with the combination at 20 µg/mL significantly enhanced SOD and GSH-Px activities compared to the H_2_O_2_ and vehicle group. Although the lower dose did not significantly affect CAT activity, a trend toward increased enzyme activity was observed.

### 3.7. Effects of Curcumin-Enriched Turmeric Extract Combined with L-Ascorbic Acid on Inflammatory Markers

Inflammation plays a crucial role in the progression of neurodegenerative diseases. The transcription factor NF-κB regulates inflammatory responses, including the expression of IL-6, a pro-inflammatory cytokine. IL-6 can further activate NF-κB, creating a feedback loop that exacerbates inflammation.

To assess the effects of curcumin-enriched turmeric extract combined with L-ascorbic acid on these inflammatory markers, we measured NF-κB and IL-6 levels in H_2_O_2_-induced SH-SY5Y cells. As shown in Figure 7, cells treated with H_2_O_2_ and vehicle exhibited significantly higher levels of NF-κB and IL-6 (*p* < 0.01 and *p* < 0.001, respectively, compared to the control group). Notably, treatment with the combination significantly reduced these elevated inflammatory markers (*p* < 0.05 for 20 µg/mL; *p* < 0.01 for 40 µg/mL) compared to the H_2_O_2_-plus-vehicle group.

### 3.8. Effects of Curcumin-Enriched Turmeric Extract Combined with L-Ascorbic Acid on Apoptotic Markers

Caspase-3, a key executioner caspase, mediates cell death, while caspase-9 functions as an initiator caspase, activating caspase-3 within the intrinsic apoptotic pathway. Bcl-2, an anti-apoptotic protein, helps protect cells by preventing mitochondrial damage. Together, these molecules regulate apoptosis, ensuring a balance between cell survival and death.

In this study, we evaluated the effects of curcumin-enriched turmeric extract combined with L-ascorbic acid on apoptosis in H_2_O_2_-treated SH-SY5Y cells. Figure 8 shows that cells exposed to H_2_O_2_ and vehicle exhibited a significant reduction in Bcl-2 expression (*p* < 0.01, compared to the control), alongside a marked increase in the levels of both caspase-3 (*p* < 0.001, compared to the control) and caspase-9 (*p* < 0.05, compared to the control).

Importantly, treatment with the combination of curcumin-enriched turmeric extract and L-ascorbic acid at doses of 20 µg/mL and 40 µg/mL notably restored Bcl-2 levels (*p* < 0.05 and *p* < 0.01, respectively, compared to the H_2_O_2_ + vehicle group). Additionally, the combination significantly reduced caspase-3 and caspase-9 expression (*p* < 0.001 for all doses of caspase-3 and *p* < 0.05 for all doses of caspase-9, compared to the H_2_O_2_ + vehicle group).

## 4. Discussion

This study investigated the neuroprotective effects of combining curcumin-enriched turmeric extract with L-ascorbic acid to mitigate oxidative damage induced by H_2_O_2_ in SH-SY5Y neuroblastoma cells. The results demonstrate that this combination significantly enhances neuronal survival by targeting oxidative stress, inflammation, and undesired apoptosis, with a key focus on epigenetic modulation. The synergistic interactions between curcumin and L-ascorbic acid are central to their combined neuroprotective efficacy, offering a promising strategy for neurodegenerative disease management.

The principle of synergism, where the combined effects of two or more agents exceed the sum of their individual effects, is fundamental in pharmacology and plays a critical role in optimizing therapeutic strategies [26,27]. This study provides compelling evidence that the combination of curcumin and L-ascorbic acid exhibits synergistic neuroprotective properties, particularly in mitigating oxidative stress and inflammation. The enhanced efficacy of the combination is reflected in the significantly lower half-maximal effective concentration (EC50) values for antioxidant activities and COX-2 inhibition compared to the individual compounds. This suggests that curcumin and L-ascorbic acid interact at multiple molecular targets, amplifying their biological effects beyond what each compound can achieve alone.

The confirmation of synergy through combination index (CI) values (<1) in FRAP, DPPH, and ABTS antioxidant assays, along with COX-2 suppression activity, further supports the hypothesis that these compounds work through complementary mechanisms. Curcumin, a polyphenolic compound, is known for its ability to neutralize ROS and modulate redox-sensitive transcription factors, while L-ascorbic acid functions as a direct ROS scavenger and a cofactor in enzymatic antioxidant reactions. The interplay between these mechanisms likely enhances the stabilization of redox homeostasis, leading to superior protection against oxidative insults.

Additionally, the observed dose reduction index (>1) suggests that lower doses of the combination can achieve the same therapeutic effects as higher doses of the individual compounds. This has significant implications for clinical applications, as it may reduce the risk of dose-dependent toxicity while improving long-term safety. Curcumin, despite its potent bioactivity, has limitations such as low bioavailability and potential gastrointestinal irritation at high doses [28]. L-ascorbic acid, while well-tolerated, may contribute to pro-oxidant effects at excessive concentrations. The synergistic interaction between the two compounds allows for effective neuroprotection at reduced doses, thereby minimizing adverse effects and enhancing their feasibility as a functional ingredient for long-term use in neurodegenerative disease prevention and management.

Epigenetic mechanisms play a fundamental role in cellular homeostasis, regulating oxidative stress, inflammation, and apoptosis through modifications in gene expression without altering the genetic code. In neuronal cells, oxidative stress acts as a critical factor in neurodegeneration by triggering oxidative stress, inflammation and apoptosis, leading to neuronal loss and dysfunction [29]. Among the key epigenetic regulators, SIRT1 and DNMT1 significantly influence cell fate. SIRT1 functions as a histone deacetylase that enhances cellular survival by reducing oxidative stress and inflammation, whereas DNMT1 maintains DNA methylation patterns that can either repress or activate specific gene expression pathways [7,30,31].

Our study demonstrates that curcumin-enriched turmeric extract combined with L-ascorbic acid mitigates oxidative stress by upregulating SIRT1 and downregulating DNMT1. This epigenetic modulation corresponds with a significant reduction in intracellular ROS levels, supporting the hypothesis that SIRT1 activation strengthens cellular antioxidant defenses. Notably, this combination treatment enhances the activity of key antioxidant enzymes—SOD, CAT, and GSH-Px—while concurrently lowering MDA levels, a critical marker of lipid peroxidation. These findings highlight the protective potential of this formulation in counteracting oxidative damage and preserving cellular function.

Oxidative stress, driven by excessive ROS accumulation, disrupts redox homeostasis and inflicts damage on essential biomolecules, including lipids, proteins, and DNA. In response, cells activate antioxidant defense systems to restore equilibrium [32]. Our results indicate that treatment with curcumin-enriched turmeric extract and L-ascorbic acid significantly upregulates both the expression and activity of SOD, CAT, and GSH-Px, three pivotal enzymes that collectively detoxify superoxide radicals, hydrogen peroxide, and lipid peroxides. This enzymatic enhancement aligns with SIRT1 activation, consistent with prior studies demonstrating that SIRT1 deacetylates and activates FOXO3a, a key transcription factor regulating antioxidant gene expression [33]. The resultant decline in ROS levels suggests that the observed protective effects are mediated through the epigenetic regulation of multiple antioxidant pathways, reinforcing the role of this combination in shielding neuronal cells from oxidative damage.

Conversely, DNMT1, a DNA methyltransferase, has been implicated in oxidative stress-induced epigenetic alterations that contribute to neuronal dysfunction [34,35]. Our results demonstrate that the combined treatment of curcumin-enriched turmeric extract and L-ascorbic acid significantly downregulated DNMT1 expression. This downregulation may contribute to the observed improvements in redox balance, potentially by enhancing the expression of antioxidant genes. Although the exact molecular pathways remain to be fully elucidated, the reduction in DNMT1 levels following this treatment could be attributed to a direct action of curcumin and L-ascorbic acid, as well as a secondary effect due to reduced oxidative stress. Elevated DNMT1 activity is associated with the hypermethylation of genes encoding antioxidant proteins, resulting in their transcriptional silencing [36]. Therefore, the observed reduction in DNMT1 expression may facilitate the reactivation of these protective genes, thereby reinforcing the neuroprotective effects through epigenetic modulation. 

Chronic oxidative stress is a well-established trigger for inflammation, activating intracellular signaling pathways that upregulate pro-inflammatory mediators such as NF-κB and interleukins. NF-κB serves as a transcriptional regulator of inflammatory cytokines, including IL-6, which plays a central role in neuroinflammatory responses [37,38,39]. In our study, curcumin-enriched turmeric extract combined with L-ascorbic acid treatment significantly reduced NF-κB and IL-6 expression, indicating that epigenetic modulation not only alleviates oxidative stress, but also suppresses downstream inflammatory cascades.

The observed reduction in inflammation can be attributed to SIRT1 activation, which has been shown to deacetylate NF-κB (p65 subunit), thereby inhibiting its transcriptional activity [40]. As a result, NF-κB-dependent inflammatory gene expression is suppressed, leading to decreased IL-6 levels. Given that IL-6 is a key mediator of neuroinflammation and has been implicated in neuronal damage, its reduction suggests a protective role of curcumin-enriched turmeric extract combined with L-ascorbic acid treatment in preventing inflammatory damage in SH-SY5Y cells. Furthermore, the downregulation of DNMT1 may also contribute to the suppression of inflammation, as DNMT1-mediated hypermethylation of anti-inflammatory genes can lead to their silencing. By reducing DNMT1 expression, this combination treatment may help restore the balance between pro- and anti-inflammatory signaling pathways, thereby mitigating neuroinflammatory responses. 

Prolonged oxidative stress and inflammation trigger apoptotic pathways, contributing to neuronal loss. Apoptosis is tightly regulated by the balance between pro-survival and pro-apoptotic proteins, including Bcl-2, caspase-3, and caspase-9. Bcl-2, an anti-apoptotic protein, stabilizes the mitochondrial membrane by preventing the activation of pro-apoptotic proteins like Bax and Bak, thereby inhibiting mitochondrial outer membrane permeabilization (MOMP) and the subsequent release of cytochrome c [41,42]. In contrast, caspase-9 acts as an initiator caspase, becoming activated upon cytochrome c release and forming the apoptosome complex, which then triggers the cleavage of caspase-3. Caspase-3, as an executioner caspase, degrades structural and nuclear proteins, leading to cellular dismantling and apoptosis. The interplay between these proteins determines cell fate, with Bcl-2 counteracting apoptotic signaling, while caspase-9 and caspase-3 drive the progression of programmed cell death [42,43]. 

Our results demonstrate that curcumin-enriched turmeric extract combined with L-ascorbic acid treatment significantly increased Bcl-2 expression while reducing caspase-3 and caspase-9 levels, suggesting a protective effect against oxidative stress-induced apoptosis. This anti-apoptotic response is closely linked to epigenetic modulation, as SIRT1 activation has been shown to deacetylate and inhibit p53 activity, thereby preventing the transcription of pro-apoptotic genes. Additionally, SIRT1-mediated deacetylation of FOXO3a reduces its ability to promote apoptotic gene expression, further contributing to cell survival [44,45].

The downregulation of DNMT1 may also play a role in the observed anti-apoptotic effects. Excessive DNA methylation mediated by DNMT1 has been reported to suppress the expression of pro-survival genes, shifting the balance toward apoptosis [46,47]. By decreasing DNMT1 levels, curcumin-enriched turmeric extract combined with L-ascorbic acid treatment may facilitate the re-expression of genes involved in neuronal survival, thereby preventing oxidative stress-induced cell death. 

The cumulative impact of epigenetic modulation on oxidative stress, inflammation, and apoptosis is reflected in enhanced cell viability and survival. Curcumin-enriched turmeric extract combined with L-ascorbic acid treatment significantly improved SH-SY5Y cell viability, indicating that the attenuation of oxidative stress, suppression of inflammation, and inhibition of apoptosis collectively contribute to neuronal protection. The increase in cell viability aligns with the upregulation of SIRT1, which promotes cellular resilience against oxidative damage, and the downregulation of DNMT1, which prevents detrimental epigenetic modifications.

The results of our study revealed a dose-dependent effect of curcumin-enriched turmeric extract combined with L-ascorbic acid in SH-SY5Y cells. At a dose of 20 µg/mL, while there was some indication of neuroprotection, the effects on the key markers of oxidative stress and epigenetic modulation were not significant. Specifically, CAT activity and DNMT1 expression remained unchanged at this lower dose, suggesting that a minimum threshold concentration may be required for these mechanisms to be activated effectively. However, at the higher dose of 40 µg/mL, a more robust response was observed, with significant reductions in oxidative stress markers, pro-inflammatory cytokines, and pro-apoptotic proteins, alongside enhanced antioxidant enzyme activities (including CAT).

The lack of significant effects on CAT and DNMT1 at the 20 µg/mL dose may be attributed to the insufficient concentration of curcumin and L-ascorbic acid to trigger these specific enzymatic or epigenetic pathways. It is possible that these proteins require a higher dose to achieve their functional modulation, as observed in the 40 µg/mL dose. At higher concentrations, curcumin’s antioxidant properties and L-ascorbic acid’s synergistic effect likely work in tandem to modulate oxidative stress and inflammation more effectively, thereby activating the epigenetic regulators such as DNMT1 and enhancing antioxidant defenses, which were more evident at the elevated dose.

The dose-dependent effects observed suggest that a higher concentration of the curcumin-enriched turmeric extract combined with L-ascorbic acid provides stronger protection against cellular damage, influencing both oxidative stress and inflammation pathways as well as apoptosis regulation. Higher doses may lead to a more effective scavenging of ROS, which results in reduced lipid peroxidation, while also modulating inflammatory responses through the suppression of NF-κB and IL-6. This, in turn, likely contributes to the observed reduction in apoptotic cell death.

Despite these promising findings, our study has several limitations. The use of SH-SY5Y cells, derived from human neuroblastoma, may not fully capture the complexity of neurodegenerative conditions, particularly in vivo, where multiple cell types and systemic interactions contribute to disease progression. Additionally, the bioavailability of curcumin-enriched turmeric extract and L-ascorbic acid remains uncertain, as their pharmacokinetics under physiological conditions require further investigation. Future studies should assess their stability, metabolism, and distribution in vivo to determine their therapeutic feasibility.

While our findings suggest that epigenetic regulation plays a role in neuroprotection, the precise mechanisms underlying DNMT1 downregulation require further exploration. DNMT1 expression may be more dose-sensitive than other biomarkers, warranting a deeper investigation into its regulation within the broader epigenetic landscape. In addition to this, histone acetylation and microRNAs, which are key regulators of neuroprotection, were not the primary focus of this study. Therefore, further research should explore the potential impact of curcumin and L-ascorbic acid on histone modifications and microRNA expression, as these factors may contribute significantly to the observed neuroprotective effects.

Furthermore, although the MTT assay provided insights into short-term neuroprotection, additional cytotoxicity assays—such as long-term viability assessments—are necessary to evaluate potential chronic toxicity, sustained neuronal survival, and delayed cellular responses. The relatively short exposure duration in our study also limits the assessment of long-term effects.

Moreover, while our results highlight the therapeutic potential of this combination treatment, optimizing the dose, treatment duration, and administration strategy is crucial for translational applications. Further research, including in vivo studies and clinical trials, is essential to confirm its efficacy, safety, and long-term therapeutic potential.

In summary, our study highlights the potential of curcumin-enriched turmeric extract combined with L-ascorbic acid in alleviating oxidative stress, inflammation, and apoptosis in neurodegenerative diseases. The dose-dependent effects observed suggest that higher doses provide greater neuroprotection. A novel aspect of this research lies in the synergistic impact on epigenetic regulation, offering a dual mechanism for neuroprotection. These findings emphasize the potential of curcumin-enriched turmeric extract and L-ascorbic acid as key ingredients in developing functional foods or dietary supplements to prevent or manage neurodegenerative diseases.

## 5. Conclusions

This study highlights the novel synergistic neuroprotective effects of curcumin-enriched turmeric extract combined with L-ascorbic acid. Our findings demonstrate that this combination exerts its protective properties through multiple mechanisms, including epigenetic modulation, which contributes to reducing oxidative stress, suppressing inflammation, and preventing undesired apoptosis. These results underscore the potential of this formulation as a functional ingredient for developing neuroprotective agents targeting neurodegenerative diseases. However, further studies are needed to assess its toxicity and ensure safety before progressing to preclinical and clinical applications.

## Figures and Tables

**Figure 1 foods-14-00892-f001:**
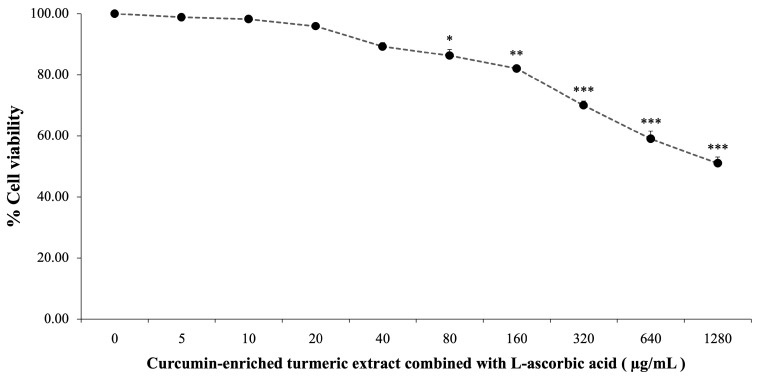
Cytotoxicity of curcumin-enriched turmeric extract combined with L-ascorbic acid on the viability of SH-SY5Y cells. Data are presented as the mean ± SEM. *, **, *** *p* < 0.05, 0.01, and 0.001, respectively; compared with the naïve control (0 µg/mL).

**Figure 2 foods-14-00892-f002:**
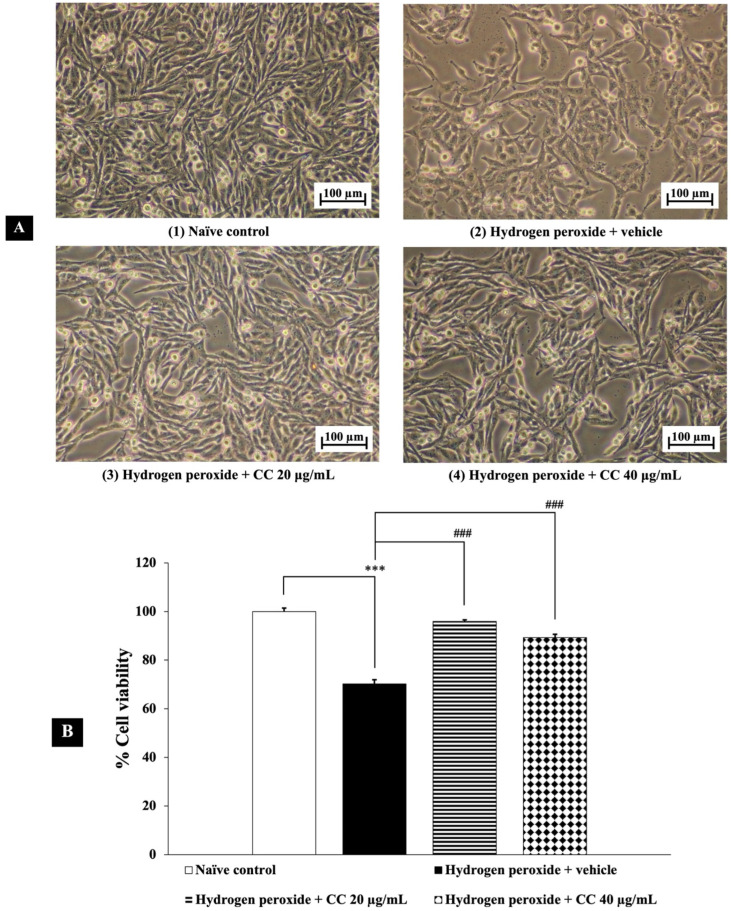
Neuroprotective effects of curcumin-enriched turmeric extract combined with L-ascorbic acid on H_2_O_2_-induced oxidative damage in SH-SY5Y cells. (**A**) Representative images of SH-SY5Y cell density at 10× magnification. (**B**) Percentage of cell viability. Data are presented as the mean ± SEM. *** *p* < 0.001; compared with the naïve control group, ^###^
*p* < 0.001; compared with the H_2_O_2_ and vehicle-treated group. H_2_O_2_: 200 µM hydrogen peroxide; CC: curcumin-enriched turmeric extract combined with L-ascorbic acid.

**Figure 3 foods-14-00892-f003:**
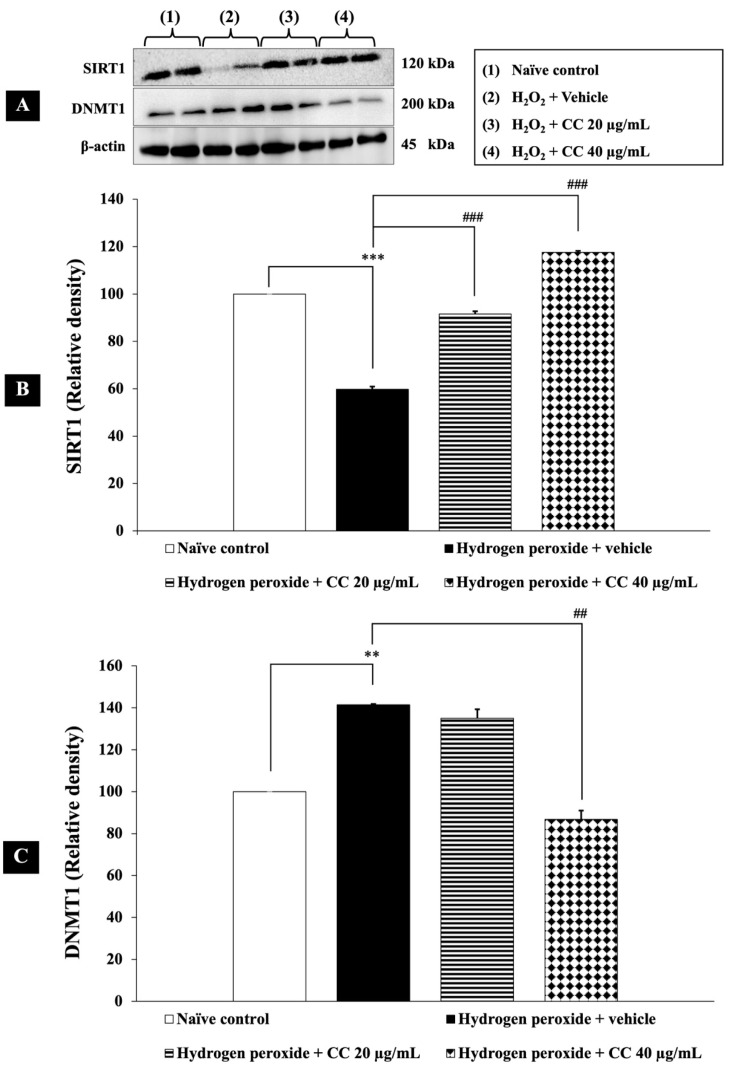
Effect of curcumin-enriched turmeric extract combined with L-ascorbic acid on epigenetic modulation in oxidative stress-induced SH-SY5Y cell damage. (**A**) Immunoblot showing the expression of SIRT1 and DNMT1. (**B**) Relative density of SIRT1, normalized to beta-actin. (**C**) Relative density of DNMT1, normalized to beta-actin. Data are presented as the mean ± SEM. **, *** *p* < 0.01 and 0.001, respectively; compared with the naïve control group, ^##^, ^###^
*p* < 0.01 and 0.001, respectively; compared with the H_2_O_2_ and vehicle-treated group. H_2_O_2_: 200 µM hydrogen peroxide; CC: curcumin-enriched turmeric extract combined with L-ascorbic acid.

**Figure 4 foods-14-00892-f004:**
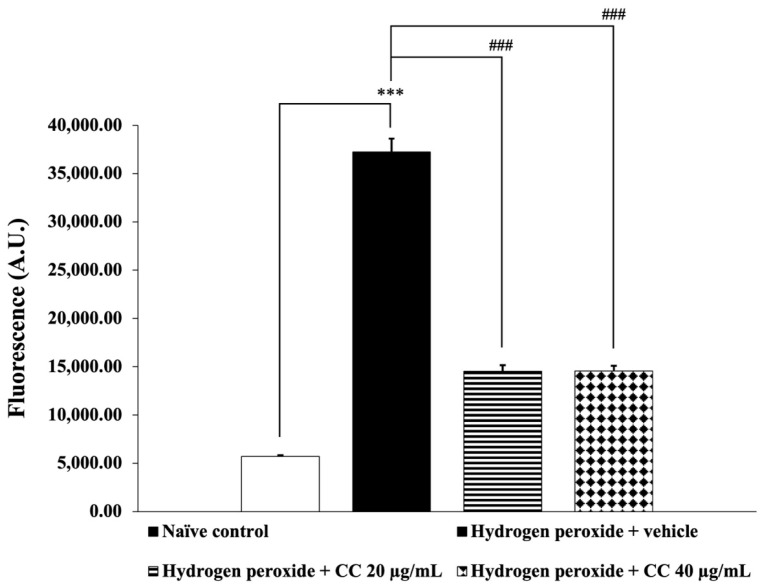
Effect of curcumin-enriched turmeric extract combined with L-ascorbic acid on ROS generation in oxidative stress-induced SH-SY5Y cell damage. Data are presented as the mean ± SEM. *** *p* < 0.001; compared with the naïve control group; ^###^
*p* < 0.001; compared with the H_2_O_2_ and vehicle-treated group. H_2_O_2_: 200 µM hydrogen peroxide; CC: curcumin-enriched turmeric extract combined with L-ascorbic acid.

**Figure 5 foods-14-00892-f005:**
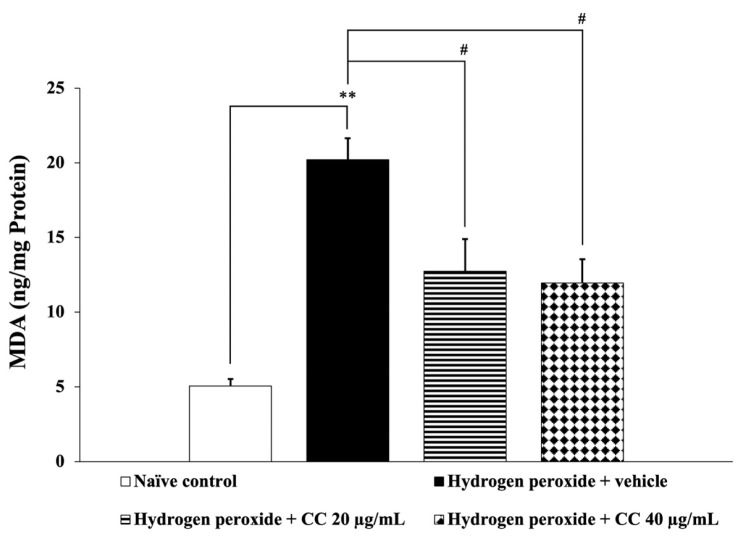
Effect of curcumin-enriched turmeric extract combined with L-ascorbic acid on MDA level in oxidative stress-induced SH-SY5Y cell damage. Data are presented as the mean ± SEM. ** *p* < 0.01; compared with the naïve control group; ^#^
*p* < 0.05; compared with the H_2_O_2_ and vehicle-treated group. H_2_O_2_: 200 µM hydrogen peroxide; CC: curcumin-enriched turmeric extract combined with L-ascorbic acid.

**Figure 6 foods-14-00892-f006:**
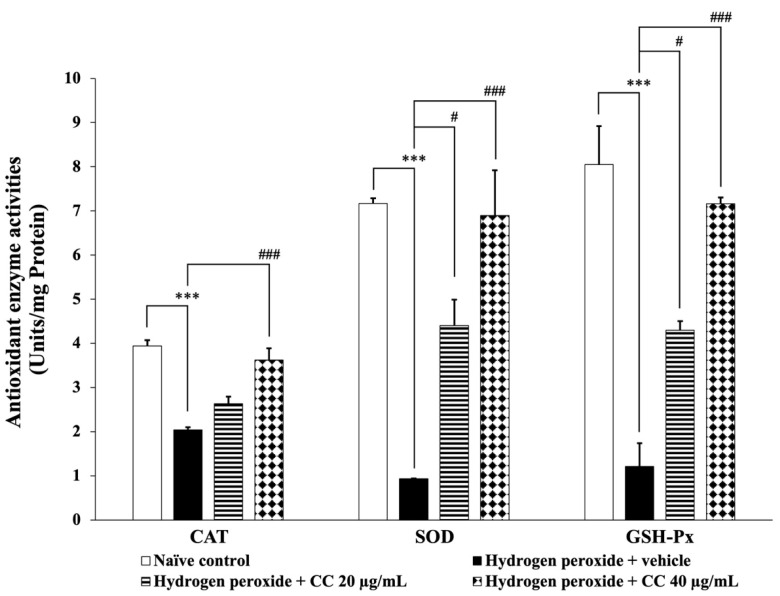
Effect of curcumin-enriched turmeric extract combined with L-ascorbic acid on antioxidant enzyme activities in oxidative stress-induced SH-SY5Y cell damage. Data are presented as the mean ± SEM. *** *p* < 0.001; compared with the naïve control group; ^#^, ^###^
*p* < 0.05 and 0.001, respectively; compared with the H_2_O_2_ and vehicle-treated group. H_2_O_2_: 200 µM hydrogen peroxide; CC: curcumin-enriched turmeric extract combined with L-ascorbic acid.

**Figure 7 foods-14-00892-f007:**
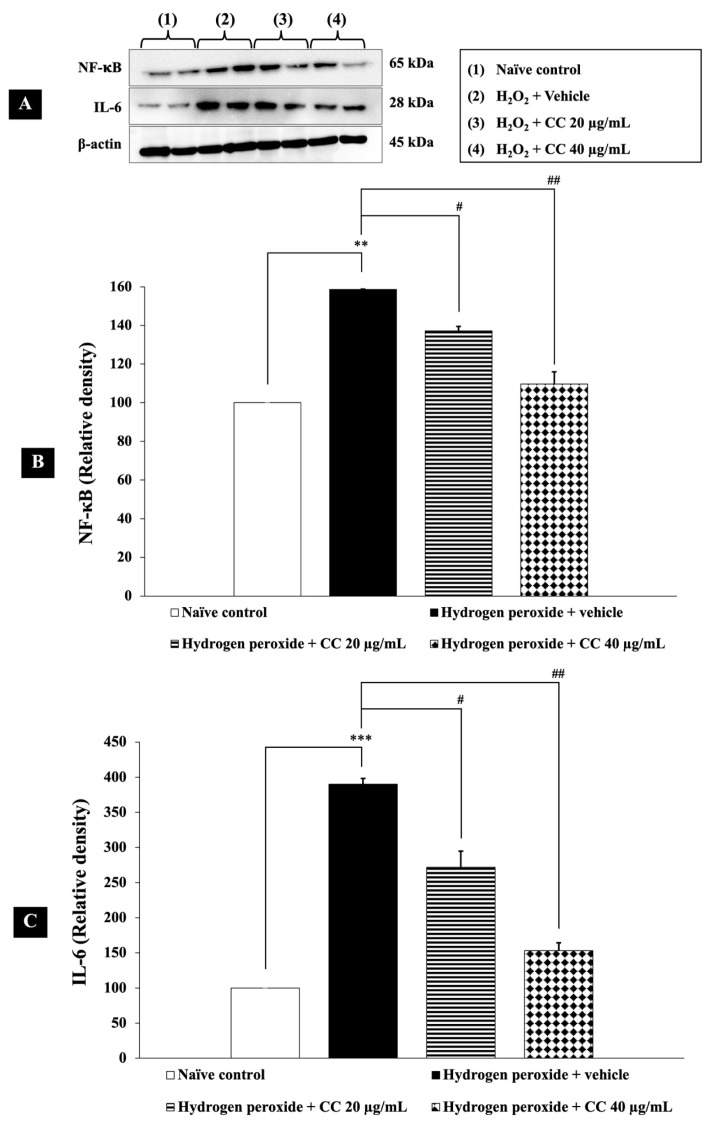
Effect of curcumin-enriched turmeric extract combined with L-ascorbic acid on inflammatory markers in oxidative stress-induced SH-SY5Y cell damage. (**A**) Immunoblot showing the expression of NF-κB and IL-6. (**B**) Relative density of NF-κB, normalized to beta-actin. (**C**) Relative density of IL-6, normalized to beta-actin. Data are presented as the mean ± SEM. **, *** *p* < 0.01 and 0.001, respectively; compared with the naïve control group, ^#^, ^##^
*p* < 0.05 and 0.01, respectively; compared with the H_2_O_2_ and vehicle-treated group. H_2_O_2_: 200 µM hydrogen peroxide; CC: curcumin-enriched turmeric extract combined with L-ascorbic acid.

**Figure 8 foods-14-00892-f008:**
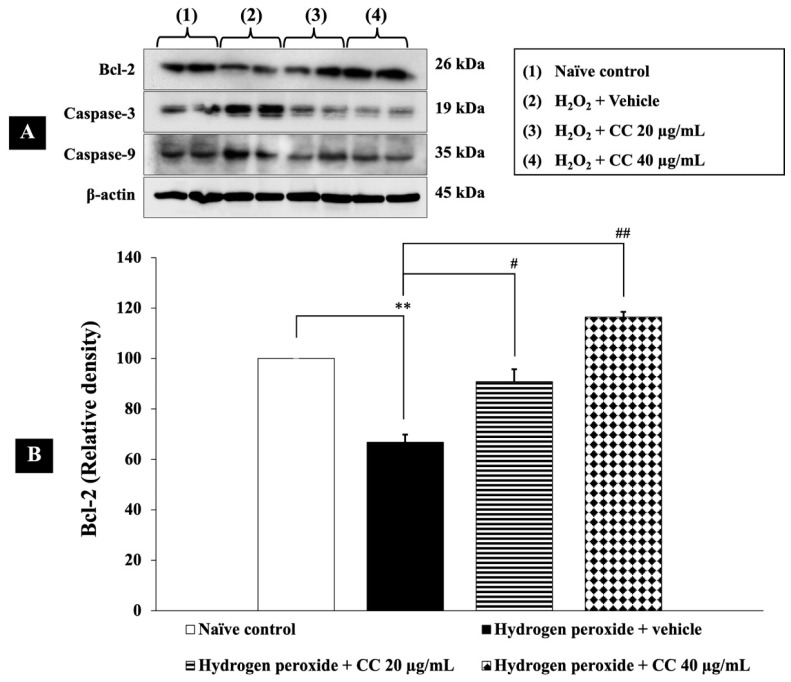
Effects of curcumin-enriched turmeric extract combined with L-ascorbic acid on apoptotic markers in H_2_O_2_-induced oxidative damage in SH-SY5Y cells. (**A**) Immunoblot showing the expression of Bcl-2, caspase-3, and caspase-9. (**B**) Relative density of Bcl-2, normalized to beta-actin. (**C**) Relative density of caspase-3, normalized to beta-actin. (**D**) Relative density of caspase-9, normalized to beta-actin. Data are presented as the mean ± SEM. *, **, *** *p* < 0.05, 0.01, and 0.001, respectively; compared with the naïve control group, ^#^, ^##^, ^###^
*p* < 0.05, 0.01, and 0.001, respectively; compared with the H_2_O_2_ and vehicle-treated group. H_2_O_2_: 200 µM hydrogen peroxide; CC: curcumin-enriched turmeric extract combined with L-ascorbic acid.

**Table 1 foods-14-00892-t001:** Synergistic effects of curcumin-enriched turmeric extract and L-ascorbic acid.

Parameter	Unit	Curcumin-Enriched Turmeric Extract	L-Ascorbic Acid	Curcumin-Enriched Turmeric Extract Combined with L-Ascorbic Acid
Antioxidant activities				
DPPH	EC_50_ (μg/mL)	67.36 ± 4.25	13.77 ± 0.18	4.70 ± 1.05 ***, ^###^
FRAP	EC_50_ (μg/mL)	26.55 ± 0.94	12.40 ± 2.35	4.64 ± 0.88 ***, ^#^
ABTS	EC_50_ (μg/mL)	75.24 ± 0.73	19.03 ± 2.62	8.06 ± 1.35 ***, ^###^
Anti-inflammatory activity				
COX-II	EC_50_ (μg/mL)	77.63 ± 0.04	121.18 ± 0.12	33.82 ± 0.05 ***, ^###^

Data are presented as the mean ± SEM. *** *p* < 0.001; comparison between curcumin-enriched turmeric extract and the combination of curcumin-enriched turmeric extract with L-ascorbic acid. ^#^, ^###^
*p* < 0.05 and 0.001, respectively; comparison between L-ascorbic acid and the combination of curcumin-enriched turmeric extract with L-ascorbic acid.

**Table 2 foods-14-00892-t002:** Combination index (CI) and dose reduction index (DRI) values of curcumin-enriched turmeric extract combined with L-ascorbic acid.

Parameter	Combination Index (Type of Interaction)	Dose Reduction Index	
		Curcumin-Enriched Turmeric Extract	L-Ascorbic Acid
Antioxidant activities			
DPPH	0.41 ± 0.05 (synergism)	15.59 ± 1.68	3.20 ± 0.35
FRAP	0.55 ± 0.02 (synergism)	6.20 ± 0.73	2.67 ± 0.00
ABTS	0.54 ± 0.06 (synergism)	9.88 ± 0.94	2.46 ± 0.25
Anti-inflammatory activity			
COX-II	0.71 ± 0.00 (synergism)	2.30 ± 0.00	3.58 ± 0.00

Data are presented as the mean ± SEM.

## Data Availability

The original contributions presented in this study are included in the article and Appendix A. Further inquiries can be directed to the corresponding authors.

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
