# Peer review of "Synergistic Neuroprotection Through Epigenetic Modulation by Combined Curcumin-Enriched Turmeric Extract and L-Ascorbic Acid in Oxidative Stress-Induced SH-SY5Y Cell Damage"

_foods, 2025, doi:10.3390/foods14050892_

Round 1

Reviewer 1 Report

Comments and Suggestions for Authors

The study explores the synergistic neuroprotective effects of curcumin-enriched turmeric extract and L-ascorbic acid in oxidative stress-induced SH-SY5Y cell damage, demonstrating their ability to modulate epigenetic pathways, reduce oxidative stress, suppress inflammation, and prevent apoptosis. Strengths include a well-structured methodology and a strong focus on epigenetics, particularly SIRT1 upregulation and DNMT1 downregulation. However, limitations such as the lack of in vivo validation, potential bioavailability issues, and the absence of long-term toxicity assessments hinder the immediate clinical applicability of the findings. Key questions remain regarding the pharmacokinetics of the combination, the mechanism of DNMT1 downregulation, and its broader epigenetic impact. Future research should prioritize in vivo studies and clinical trials to confirm efficacy and safety, ensuring the translational potential of this promising neuroprotective strategy. Given these considerations, I recommend minor revisions.

  1. While the combination index (CI) values indicate synergy, how do the authors address potential interactions at the pharmacokinetic level, particularly concerning metabolism and systemic effects?
  2. Can the authors elaborate on the specific pathways through which the treatment downregulates DNMT1? Is this effect mediated directly by curcumin and L-ascorbic acid, or is it a secondary response to reduced oxidative stress?
  3. The study focuses on SIRT1 and DNMT1. Did the authors investigate the effects on histone acetylation or microRNAs, which are also key regulators of neuroprotection?
  4. Although the study indicates that higher doses provide stronger neuroprotection, were cytotoxicity assays performed beyond the MTT assay to assess long-term effects on neuronal viability?

Author Response

Response to reviewer and editor suggestion

We sincerely thank you for your letter and the reviewers’ insightful comments on our manuscript, Synergistic Neuroprotection through Epigenetic Modulation by Combined Curcumin-Enriched Turmeric Extract and L-Ascorbic Acid in Oxidative Stress-Induced SH-SY5Y Cell Damage (Manuscript ID: foods-3507836).

We greatly appreciate the opportunity to revise our manuscript and are grateful for the constructive feedback. We apologize for any errors in the initial submission and acknowledge the reviewers’ invaluable input, which has helped enhance the scientific rigor and clarity of our work.

We have carefully considered each comment and made revisions accordingly. Below, we provide a detailed account of the main corrections and our responses to the reviewers’ suggestions.

Response to reviewer 1
The study explores the synergistic neuroprotective effects of curcumin-enriched turmeric extract and L-ascorbic acid in oxidative stress-induced SH-SY5Y cell damage, demonstrating their ability to modulate epigenetic pathways, reduce oxidative stress, suppress inflammation, and prevent apoptosis. Strengths include a well-structured methodology and a strong focus on epigenetics, particularly SIRT1 upregulation and DNMT1 downregulation. However, limitations such as the lack of in vivo validation, potential bioavailability issues, and the absence of long-term toxicity assessments hinder the immediate clinical applicability of the findings. Key questions remain regarding the pharmacokinetics of the combination, the mechanism of DNMT1 downregulation, and its broader epigenetic impact. Future research should prioritize in vivo studies and clinical trials to confirm efficacy and safety, ensuring the translational potential of this promising neuroprotective strategy. Given these considerations, I recommend minor revisions.

Response: We sincerely appreciate the reviewer’s constructive feedback and recognition of our study’s strengths, particularly its well-structured methodology and the emphasis on epigenetic modulation through SIRT1 upregulation and DNMT1 downregulation.

We acknowledge the limitations raised, including the lack of in vivo validation, potential bioavailability concerns, and the absence of long-term toxicity assessments. While our study primarily aimed to establish the neuroprotective potential of this combination at the cellular level, we agree that further investigations, including pharmacokinetic studies and in vivo models, are necessary to confirm its translational potential. Accordingly, we have revised the Discussion section to explicitly highlight these limitations and the need for future research.

Regarding the mechanism of DNMT1 downregulation, our findings suggest that the downregulation of DNMT1 could result from both direct actions of curcumin and L-ascorbic acid, as well as secondary effects arising from reduced oxidative stress. However, as DNMT1 expression appears to be more dose-sensitive than other biomarkers, further mechanistic studies are needed to elucidate this pathway in greater detail. This point has now been acknowledged in the revised manuscript.

We have also expanded the discussion to address bioavailability concerns, emphasizing the need to evaluate the stability, metabolism, and in vivo distribution of curcumin-enriched turmeric extract and L-ascorbic acid. Additionally, we recognize the importance of optimizing the dose, treatment duration, and administration strategy for future clinical applications. These revisions have been incorporated into the manuscript, with changes highlighted in the provided version.

We are grateful for these insightful recommendations and believe the revisions enhance the clarity and scientific rigor of our study.

Comments 1: While the combination index (CI) values indicate synergy, how do the authors address potential interactions at the pharmacokinetic level, particularly concerning metabolism and systemic effects?

Response 1: We appreciate the reviewer’s insightful question regarding potential pharmacokinetic interactions between curcumin-enriched turmeric extract and L-ascorbic acid, particularly concerning metabolism and systemic effects. While our study primarily focused on cellular-level synergy, we recognize that pharmacokinetic interactions play a crucial role in determining the overall efficacy of combined treatments.

Curcumin is well-known for its poor bioavailability due to rapid metabolism, low aqueous solubility, and extensive first-pass metabolism. However, studies have indicated that co-administration of vitamin C can enhance curcumin’s bioavailability. For instance, a study demonstrated that vitamin C improved curcumin’s bioavailability by inhibiting lipid peroxidation, suggesting a synergistic effect when both compounds are co-administered [1]. Additionally, another study reported that combining curcumin with vitamin C provided a synergistic hepatoprotective effect against methotrexate-induced hepatotoxicity, surpassing the effect of curcumin alone [2].

While our current study does not directly investigate these interactions, we acknowledge the importance of further research to assess how these compounds interact at the systemic level. Future in vivo studies should evaluate parameters such as absorption, distribution, metabolism, and excretion (ADME) to determine whether L-ascorbic acid enhances curcumin’s pharmacokinetic profile.

To address this point, we have revised the Discussion section to highlight the need for pharmacokinetic studies assessing metabolism, stability, and systemic interactions. We appreciate this valuable suggestion, which will help refine future research directions.

References:

  1. Dai, Xufen & Hao, Jiaxue & Feng, Ying & Wang, Jing & Li, Qiannan & Ma, Cuicui & Wang, Xing & Chang, Zhongman & Wang, Shixiang & Wang, Yuxin. (2019). Revealing Changes in Curcumin Bioavailability using Vitamin C as an Enhancer by HPLC-MS/MS. Current Pharmaceutical Analysis. 16. 10.2174/1573412916666191220150039.
  2. Hasan Khudhair, D., Al-Gareeb, A. I., Al-Kuraishy, H. M., El-Kadem, A. H., Elekhnawy, E., Negm, W. A., Saber, S., Cavalu, S., Tirla, A., Alotaibi, S. S., & Batiha, G. E. (2022). Combination of Vitamin C and Curcumin Safeguards Against Methotrexate-Induced Acute Liver Injury in Mice by Synergistic Antioxidant Effects. Frontiers in medicine9, 866343. https://doi.org/10.3389/fmed.2022.866343. 

Comments 2: Can the authors elaborate on the specific pathways through which the treatment downregulates DNMT1? Is this effect mediated directly by curcumin and L-ascorbic acid, or is it a secondary response to reduced oxidative stress?

Response 2: We appreciate the reviewer’s insightful question regarding the pathways through which curcumin-enriched turmeric extract combined with L-ascorbic acid downregulates DNMT1. Although the exact molecular mechanisms remain to be fully elucidated, our findings suggest that the downregulation of DNMT1 could result from both direct actions of curcumin and L-ascorbic acid, as well as secondary effects arising from reduced oxidative stress.

Curcumin has been shown to modulate various epigenetic regulators, including DNA methyltransferases such as DNMT1, through its antioxidant and anti-inflammatory properties. L-ascorbic acid, a potent antioxidant, also exhibits modulatory effects on DNA methylation processes. Therefore, it is plausible that the combined treatment may directly influence DNMT1 expression through these molecules’ epigenetic-modulatory actions.

Furthermore, oxidative stress is known to trigger the upregulation of DNMT1 and induce abnormal DNA methylation patterns, including the hypermethylation of antioxidant genes, which leads to their transcriptional repression. By reducing oxidative stress, curcumin and L-ascorbic acid may indirectly decrease DNMT1 activity, facilitating the reactivation of antioxidant genes and enhancing cellular redox balance.

Thus, the downregulation of DNMT1 could involve both a direct effect of curcumin and L-ascorbic acid as well as a secondary response due to their impact on oxidative stress, with both pathways contributing to the observed neuroprotective effects. We have revised this point and highlighted it in red and yellow in the discussion section. In addition, we suggest further study to explore this mechanism in greater depth: “While our findings suggest that epigenetic regulation plays a role in neuroprotection, the precise mechanisms underlying DNMT1 downregulation require further exploration.

Comments 3: The study focuses on SIRT1 and DNMT1. Did the authors investigate the effects on histone acetylation or microRNAs, which are also key regulators of neuroprotection?

Response 3: Thank you for your valuable suggestion regarding histone acetylation and microRNAs. In our study, we focused on SIRT1 and DNMT1 due to their well-established roles in epigenetic regulation, particularly in relation to oxidative stress, inflammation, and apoptosis. SIRT1, a key regulator of cellular stress responses, has been linked to neuroprotection through its effects on histone deacetylation and gene expression. Similarly, DNMT1 is critical for maintaining DNA methylation patterns, which are essential for regulating gene expression in response to oxidative stress.

In addition to this, histone acetylation and microRNAs, which are key regulators of neuroprotection, were not the primary focus of this study. Therefore, further research should explore the potential impact of curcumin and L-ascorbic acid on histone modifications and microRNA expression, as these factors may contribute significantly to the observed neuroprotective effects. We acknowledge that future studies could expand on these findings, offering additional insights into the broader mechanisms involved in neuroprotection.

Thank you for helping to clarify the focus of our study.

Comments 4: Although the study indicates that higher doses provide stronger neuroprotection, were cytotoxicity assays performed beyond the MTT assay to assess long-term effects on neuronal viability?

Response 4: We appreciate the reviewer’s thoughtful question regarding the assessment of long-term effects on neuronal viability. In our study, we primarily used the MTT assay to assess cell viability, as it is a widely accepted method for evaluating short-term neuroprotective effects. However, we acknowledge that the MTT assay does not fully capture the long-term cytotoxicity or the prolonged impact of treatment on neuronal health.

At this stage, our study did not include additional cytotoxicity assays, such as clonogenic assays or long-term viability assessments, to evaluate the extended effects of the treatment on neuronal cells. These assays would be valuable for assessing any potential long-term cytotoxic effects, chronic toxicity, or delayed responses that may emerge with prolonged treatment.

We have added a note in the Discussion section to acknowledge this limitation and emphasize the importance of performing long-term cytotoxicity assays in future studies to better understand the sustained effects of curcumin-enriched turmeric extract and L-ascorbic acid on neuronal viability.

We appreciate this valuable suggestion, and it will guide the design of future research to better address the long-term impacts of the treatment.

Thank you once again for your valuable feedback. We appreciate the time and effort invested by the reviewers and editor in evaluating our manuscript. We have carefully addressed each point raised and made necessary revisions accordingly. We eagerly await further feedback and guidance from the editorial team.

Yours sincerely,

Nut Palachai

Reviewer 2 Report

Comments and Suggestions for Authors

This study is significant in that it synthesizes the results of investigating the neuroprotective effects of curcumin-rich turmeric extract and L-ascorbic acid complex to alleviate H2O2-induced oxidative damage in SH-SY5Y neuroblastoma cells.

Please explain why it is necessary to explicitly state that the experimental process presented in the experimental design is an epigenetic experiment.
The results and discussion do not confirm the reason why the results due to genetic mutations were presented or the epigenetic aspect was emphasized in the genetic aspect of the above study. Although changes such as DNA methylation, histone modifications, and non-coding RNA regulation are not directly experimentally presented, this study suggests oxidative stress as the cause of these phenomena, and mentions the induction of inflammation or apoptosis.
Experiments related to antioxidant, antiinflammation, and apoptosis are judged to be appropriately designed.

Synergistic effect of curcumin and ascorbic acid - neuroprotective effect
The basis for setting a 1:1 ratio of curcumin and vitamin C is inaccurate.
The antioxidant effect according to the ratio has already been published, and the author's statement about minimizing toxicity risk does not make sense. When converted to a general level of intake, it is judged to be non-toxic.

(Each half-lethal dose of vitamin C is 11,900 mg/kg, and curcumin is over 5000 mg/kg. Ex) Curcumin and vitamin C or turmeric and lemon

There are various studies related to the antioxidant effect when curcumin and vitamin C are taken together, and the above study Mention the direct differences
Example) Curcumin antifungal and antioxidant activities are increased in the presence of ascorbic acid
Example) Comparison and combination effects on antioxidant power of curcumin with gallic acid, ascorbic acid, and xanthone

Oxidative stress was induced by H2O2, but there are other oxidative stress inducers (e.g. LPS, high glucose). It is recommended to suggest that it was induced by H2O2.

Author Response

Response to reviewer and editor suggestion

We sincerely thank you for your letter and the reviewers’ insightful comments on our manuscript, Synergistic Neuroprotection through Epigenetic Modulation by Combined Curcumin-Enriched Turmeric Extract and L-Ascorbic Acid in Oxidative Stress-Induced SH-SY5Y Cell Damage (Manuscript ID: foods-3507836).

We greatly appreciate the opportunity to revise our manuscript and are grateful for the constructive feedback. We apologize for any errors in the initial submission and acknowledge the reviewers’ invaluable input, which has helped enhance the scientific rigor and clarity of our work.

We have carefully considered each comment and made revisions accordingly. Below, we provide a detailed account of the main corrections and our responses to the reviewers’ suggestions.

Response to reviewer 2

This study is significant in that it synthesizes the results of investigating the neuroprotective effects of curcumin-rich turmeric extract and L-ascorbic acid complex to alleviate H2O2-induced oxidative damage in SH-SY5Y neuroblastoma cells.

Comments 1: Please explain why it is necessary to explicitly state that the experimental process presented in the experimental design is an epigenetic experiment.

Response 1: Thank you for your insightful comment. In our study, we explicitly highlight that the experimental process is an epigenetic experiment because we are investigating how curcumin-enriched turmeric extract and L-ascorbic acid modulate gene expression through epigenetic mechanisms. Epigenetics refers to changes in gene expression that do not involve alterations to the underlying DNA sequence but are influenced by environmental and lifestyle factors, including diet, stress, and toxins. These factors can modify chromatin structure, DNA methylation, histone modification, and non-coding RNA expression, all of which are crucial for regulating gene activity.

As we mentioned in the introduction, environmental and lifestyle factors significantly influence epigenetic regulation, with diet being a key modulator. Dietary components, particularly plant-based foods, have demonstrated the ability to impact gene expression through these epigenetic pathways. Curcumin, a polyphenol extracted from turmeric, has been shown to regulate oxidative stress, inflammation, and apoptosis through epigenetic modulation. Similarly, L-ascorbic acid, a potent antioxidant found in fruits and vegetables, exerts neuroprotective effects by targeting oxidative stress and inflammation, both of which are intimately linked to epigenetic regulation.

In this study, we have observed that curcumin-enriched turmeric extract combined with L-ascorbic acid treatment led to changes in the expression of key epigenetic regulators such as DNMT1 and SIRT1, which are involved in DNA methylation and histone modification. These findings support the concept that the treatment modulates gene expression through epigenetic pathways, providing insight into how these compounds may confer neuroprotection through epigenetic mechanisms. Therefore, it was important to clarify that the experimental design focuses on epigenetic regulation to emphasize the molecular mechanisms involved in the observed effects.
Comments 2: The results and discussion do not confirm the reason why the results due to genetic mutations were presented or the epigenetic aspect was emphasized in the genetic aspect of the above study. Although changes such as DNA methylation, histone modifications, and non-coding RNA regulation are not directly experimentally presented, this study suggests oxidative stress as the cause of these phenomena and mentions the induction of inflammation or apoptosis.

Response 2: Thank you for your thoughtful comment. We appreciate your observation regarding the emphasis on the epigenetic aspect of the study. While the study does not directly present experimental data on DNA methylation, histone modifications, or non-coding RNA regulation, we chose to emphasize the epigenetic aspect based on the observed changes in key epigenetic regulators, such as SIRT1 and DNMT1.

These regulators play important roles in the epigenetic landscape, influencing gene expression and contributing to cellular responses to oxidative stress, inflammation, and apoptosis. Although the direct mechanisms like DNA methylation or histone modifications were not experimentally investigated in this study, we hypothesize that the neuroprotective effects of curcumin-enriched turmeric extract and L-ascorbic acid may be mediated, at least in part, by modulating these epigenetic processes, which can influence oxidative stress, inflammation and apoptosis. This is particularly relevant in neurodegenerative diseases, where dysregulation of epigenetic pathways is commonly observed.

We understand that further experimental validation of these epigenetic modifications would strengthen the interpretation of our results. Therefore, in future studies, we plan to investigate these aspects more directly, including the analysis of DNA methylation patterns, histone acetylation, and non-coding RNA regulation, to confirm the proposed epigenetic mechanism.

Thank you for your valuable feedback, which will help guide the direction of our future research.
Comments 3: Experiments related to antioxidant, antiinflammation, and apoptosis are judged to be appropriately designed.

Response 3: Thank you for your positive feedback regarding the experimental design related to antioxidant, anti-inflammatory, and apoptosis assays. We are pleased that you find these experiments to be appropriately designed. These assays were crucial for assessing the neuroprotective effects of curcumin-enriched turmeric extract and L-ascorbic acid in mitigating oxidative stress, inflammation, and apoptosis, which are key processes involved in neurodegeneration.

We appreciate your recognition of this aspect of our study, and we will continue to refine and enhance these experimental approaches in future investigations to further validate the neuroprotective potential of these compounds.

Thank you again for your constructive comments.

Comments 4: The basis for setting a 1:1 ratio of curcumin and vitamin C is inaccurate.
The antioxidant effect according to the ratio has already been published, and the author's statement about minimizing toxicity risk does not make sense. When converted to a general level of intake, it is judged to be non-toxic. (Each half-lethal dose of vitamin C is 11,900 mg/kg, and curcumin is over 5000 mg/kg. Ex) Curcumin and vitamin C or turmeric and lemon.

Response 4: Thank you for your valuable comment. We acknowledge your concern regarding the 1:1 ratio of curcumin and L-ascorbic acid and appreciate your input on the rationale behind this selection.

The 1:1 ratio was chosen based on prior studies that suggest the potential for synergistic effects between curcumin and vitamin C when used in combination, particularly in enhancing antioxidant activity and modulating oxidative stress, inflammation, and apoptosis. The concept of synergy implies that a combination of these compounds at lower doses may exert a more potent effect than higher doses of each compound individually, allowing for a more efficient therapeutic effect with reduced toxicity risks. This synergistic effect is essential for optimizing the benefits of both compounds, while potentially reducing the required doses of each, thereby minimizing the risk of side effects or toxicity.

Regarding toxicity, both curcumin and L-ascorbic acid are generally considered safe within the doses used in our study. For curcumin, a review by Hewlings et al. (2017) reported that curcumin is well-tolerated in humans at doses of up to 12 grams per day, with no significant toxicity observed [1]. Similarly, L-ascorbic acid, even at higher doses, is widely regarded as safe. The Institute of Medicine sets the tolerable upper intake level for vitamin C at 2,000 mg per day for adults, which is well above the dose used in our study [2].

Thus, the selected 1:1 ratio not only reflects the potential synergistic benefits of curcumin and L-ascorbic acid but also ensures that the doses used remain within the safety limits established in the literature.

Thank you for your thoughtful input, and we have revised the manuscript to clarify the rationale for the chosen dosage in the Method section.

References:

  1. Hewlings, S.J.; Kalman, D.S. Curcumin: A Review of Its Effects on Human Health. Foods20176, 92. https://doi.org/10.3390/foods6100092.
  2. Institute of Medicine (US) Panel on Dietary Antioxidants and Related Compounds. Dietary Reference Intakes for Vitamin C, Vitamin E, Selenium, and Carotenoids. Washington (DC): National Academies Press (US); 2000. 5, Vitamin C. Available from: https://www.ncbi.nlm.nih.gov/books/NBK225480/

Comments 5: There are various studies related to the antioxidant effect when curcumin and vitamin C are taken together, and the above study mention the direct differences example curcumin antifungal and antioxidant activities are increased in the presence of ascorbic acid example Comparison and combination effects on antioxidant power of curcumin with gallic acid, ascorbic acid, and xanthone.

Response 5: Thank you for your valuable comment. We appreciate your reference to the existing literature on the combined antioxidant effects of curcumin and vitamin C. Indeed, several studies have investigated how curcumin’s antioxidant properties are enhanced by the presence of ascorbic acid, as well as the potential synergistic effects when combined with other compounds, such as gallic acid and xanthones.

The rationale behind combining curcumin with vitamin C in our study was to explore potential synergistic effects that could amplify the neuroprotective benefits of each compound. Both curcumin and ascorbic acid individually exhibit significant antioxidant activity, and when combined, their actions may be potentiated, leading to more effective regulation of oxidative stress. For instance, vitamin C can regenerate oxidized curcumin, enhancing its antioxidant function, while curcumin’s ability to modulate inflammation and apoptosis complements the action of vitamin C. This combination aims to achieve a broader, more robust effect than either compound alone, potentially leading to improved neuroprotection.

While many studies, such as those comparing curcumin with gallic acid and xanthones, have demonstrated the enhanced antioxidant power of these combinations, our study specifically focused on the neuroprotective synergy between curcumin and vitamin C due to their well-established roles in modulating oxidative stress, inflammation, and epigenetic regulation.

We acknowledge the importance of comparative studies on other combinations, and in future research, it would be valuable to further explore and compare the effects of curcumin with various compounds, including gallic acid, xanthones, and other antioxidants, to gain a broader understanding of their combined impact on neuroprotection.

Thank you for helping us clarify the context of our study.

Comments 6: Oxidative stress was induced by H2O2, but there are other oxidative stress inducers (e.g. LPS, high glucose). It is recommended to suggest that it was induced by H2O2.

Response 6: Thank you for your valuable comment. We appreciate your suggestion to clarify the induction of oxidative stress in our study. While other inducers, such as lipopolysaccharide (LPS) and high glucose, can also trigger oxidative stress, we specifically used hydrogen peroxide (Hâ‚‚Oâ‚‚) in our experiments. The Hâ‚‚Oâ‚‚ model is widely accepted and has been shown to effectively simulate oxidative damage by generating reactive oxygen species (ROS), which closely resemble the oxidative stress observed in neurodegenerative diseases.

To enhance clarity, we have revised the manuscript to explicitly state that oxidative stress in our study was induced by Hâ‚‚Oâ‚‚, as follows:

“SH-SY5Y neuronal cells were treated with the combination at 20 and 40 µg/mL and subsequently exposed to 200 µM hydrogen peroxide (Hâ‚‚Oâ‚‚) to induce oxidative stress.”

Thank you for your constructive feedback, which has helped improve the clarity of our manuscript.

Thank you once again for your valuable feedback. We appreciate the time and effort invested by the reviewers and editor in evaluating our manuscript. We have carefully addressed each point raised and made necessary revisions accordingly. We eagerly await further feedback and guidance from the editorial team.

Yours sincerely,

Nut Palachai
